# A Closer Look at Accuracy vs. Robustness

**Yao-Yuan Yang**[*1]     **Cyrus Rashtchian**[*1]     **Hongyang Zhang**[2]

**Ruslan Salakhutdinov**[3]     **Kamalika Chaudhuri**[1]

[1]University of California, San Diego
[2]Toyota Technological Institute at Chicago
[3]Carnegie Mellon University
{yay005, crashtchian}@eng.ucsd.edu   hongyanz@ttic.edu
rsalakhu@cs.cmu.edu   kamalika@cs.ucsd.edu

## Abstract

Current methods for training robust networks lead to a drop in test accuracy, which has led prior works to posit that a robustness-accuracy tradeoff may be inevitable in deep learning. We take a closer look at this phenomenon and first show that real image datasets are actually separated. With this property in mind, we then prove that robustness and accuracy should both be achievable for benchmark datasets through locally Lipschitz functions, and hence, there should be no inherent tradeoff between robustness and accuracy. Through extensive experiments with robustness methods, we argue that the gap between theory and practice arises from two limitations of current methods: either they fail to impose local Lipschitzness or they are insufficiently generalized. We explore combining dropout with robust training methods and obtain better generalization. We conclude that achieving robustness and accuracy in practice may require using methods that impose local Lipschitzness and augmenting them with deep learning generalization techniques.[1]

## 1   Introduction

A growing body of research shows that neural networks are vulnerable to *adversarial examples*, test inputs that have been modified slightly yet strategically to cause misclassification [56, 17]. While a number of defenses have been proposed [10, 32, 46, 65], they are known to hurt test accuracy on many datasets [41, 32, 68]. This observation has led prior works to claim that a tradeoff between robustness and accuracy may be *inevitable* for many classification tasks [57, 65].

We take a closer look at the tradeoff between robustness and accuracy, aiming to identify properties of data and training methods that enable neural networks to achieve *both*. A plausible reason why robustness may lead to lower accuracy is that different classes are very close together or they may even overlap (which underlies the argument for an inevitable tradeoff [57]). We begin by testing if this is the case in real data through an empirical study of four image datasets. Perhaps surprisingly, we find that these datasets actually satisfy a natural separation property that we call $r$-separation: examples from different classes are at least distance $2r$ apart in pixel space. This $r$-separation holds for values of $r$ that are higher than the perturbation radii used in adversarial example experiments.

We next consider separation as a guiding principle for better understanding the robustness-accuracy tradeoff. Neural network classifiers are typically obtained by rounding an underlying continuous

function $f : \mathcal{X} \to \mathbb{R}^C$ with $C$ classes. We take inspiration from prior work, which shows that Lipschitzness of $f$ is closely related to its robustness [10, 21, 46, 60, 64]. However, one drawback of the existing arguments is that they do not provide a compelling and realistic assumption on the data that guarantees robustness and accuracy. We show theoretically that any $r$-separated data distribution has a classifier that is both robust up to perturbations of size $r$, and accurate, and it can be obtained by rounding a function that is locally Lipschitz around the data. This suggests that there should exist a robust and highly accurate classifier for real image data. Unfortunately, the current state of robust classification falls short of this prediction, and the discrepancy remains poorly understood.

To better understand the theory-practice gap, we empirically investigate several existing methods on a few image datasets with a special focus on their local Lipschitzness and generalization gaps. We find that of the methods investigated, adversarial training (AT) [32], robust self-training (RST) [42] and TRADES [64] impose the highest degree of local smoothness, and are the most robust. We also find that the three robust methods have large gaps between training and test accuracies as well as adversarial training and test accuracies. This suggests that the disparity between theory and practice may be due to the limitations of existing training procedures, particularly in regards to generalization. We then experiment with dropout, a standard generalization technique, on top of robust training methods on two image datasets where there is a significant generalization gap. We see that dropout in particular narrows the generalization gaps of TRADES and RST, and improves test accuracy, test adversarial accuracy as well as test Lipschitzness. In summary, our contributions are as follows.

- Through empirical measurements, we show that several image datasets are separated.

- We prove that this separation implies the existence of a robust and perfectly accurate classifier that can be obtained by rounding a locally Lipschitz function. In contrast to prior conjectures [12, 16, 57], robustness and accuracy can be achieved together in principle.

- We investigate smoothness and generalization properties of classifiers produced by current training methods. We observe that the training methods AT, TRADES, and RST, which produce robust classifiers, also suffer from large generalization gaps. We combine these robust training methods with dropout [54], and show that this narrows the generalization gaps and sometimes makes the classifiers smoother.

What do our results imply about the robustness-accuracy tradeoff in deep learning? They suggest that this tradeoff is not inherent. Rather, it is a consequence of current robustness methods. The past few years of research in robust machine learning has led to a number of new loss functions, yet the rest of the training process – network topologies, optimization methods, generalization tools – remain highly tailored to promoting accuracy. We believe that in order to achieve both robustness and accuracy, future work may need to redesign other aspects of the training process such as better network architectures using neural architecture search [13, 19, 47, 69]. Combining this with improved optimization methods and robust losses may be able to reduce the generalization gap in practice.

## 2 Preliminaries

Let $\mathcal{X} \subseteq \mathbb{R}^d$ be an instance space equipped with a metric $\text{dist} : \mathcal{X} \times \mathcal{X} \to \mathbb{R}^+$; this is the metric in which robustness is measured. Let $[C] = \{1, 2, \dots, C\}$ denote the set of possible labels with $C \geq 2$. For a function $f : \mathcal{X} \to \mathbb{R}^C$, let $f(\boldsymbol{x})_i$ denote the value of the $i$th coordinate.

**Robustness and Astuteness.** Let $\mathbb{B}(\boldsymbol{x}, \varepsilon)$ denote a ball of radius $\varepsilon > 0$ around $\boldsymbol{x}$ in a metric space. We use $\mathbb{B}_\infty$ to denote the $\ell_\infty$ ball. A classifier $g$ is *robust* at $\boldsymbol{x}$ with radius $\varepsilon > 0$ if for all $\boldsymbol{x}' \in \mathbb{B}(\boldsymbol{x}, \varepsilon)$, we have $g(\boldsymbol{x}') = g(\boldsymbol{x})$. Also, $g$ is *astute* at $(\boldsymbol{x}, y)$ if $g(\boldsymbol{x}') = y$ for all $\boldsymbol{x}' \in \mathbb{B}(\boldsymbol{x}, \varepsilon)$. The *astuteness* of $g$ at radius $\varepsilon > 0$ under a distribution $\mu$ is

$$\Pr_{(\boldsymbol{x}, y) \sim \mu} [g(\boldsymbol{x}') = y \text{ for all } \boldsymbol{x}' \in \mathbb{B}(\boldsymbol{x}, \varepsilon)].$$

The goal of robust classification is to find a $g$ with the highest astuteness [59]. We sometimes use *clean accuracy* to refer to standard test accuracy (no adversarial perturbation), in order to differentiate it from *robust accuracy* a.k.a. astuteness (with adversarial perturbation).

**Local Lipschitzness.** Here we define local Lipschitzness theoretically; Section 5 later provides an empirical way to estimate this quantity.

**Definition 1.** *Let $(\mathcal{X}, \mathsf{dist})$ be a metric space. A function $f : \mathcal{X} \to \mathbb{R}^C$ is L-locally Lipschitz at radius $r$ if for each $i \in [C]$, we have $|f(\boldsymbol{x})_i - f(\boldsymbol{x}')_i| \leq L \cdot \mathsf{dist}(\boldsymbol{x}, \boldsymbol{x}')$ for all $\boldsymbol{x}'$ with $\mathsf{dist}(\boldsymbol{x}, \boldsymbol{x}') \leq r$.*

**Separation.** We formally define separated data distributions as follows. Let $\mathcal{X}$ contain $C$ disjoint classes $\mathcal{X}^{(1)}, \ldots, \mathcal{X}^{(C)}$, where all points in $\mathcal{X}^{(i)}$ have label $i$ for $i \in [C]$.

**Definition 2** (*r*-separation). *We say that a data distribution over $\bigcup_{i \in [C]} \mathcal{X}^{(i)}$ is r-separated if $\mathsf{dist}(\mathcal{X}^{(i)}, \mathcal{X}^{(j)}) \geq 2r$ for all $i \neq j$, where $\mathsf{dist}(\mathcal{X}^{(i)}, \mathcal{X}^{(j)}) = \min_{\boldsymbol{x} \in \mathcal{X}^{(i)}, \boldsymbol{x}' \in \mathcal{X}^{(j)}} \mathsf{dist}(\boldsymbol{x}, \boldsymbol{x}')$.*

In other words, the distance between any two examples from different classes is at least $2r$. One of our motivating observations is that many real classification tasks comprise of separated classes; for example, if dist is the $\ell_\infty$ norm, then images with different categories (e.g., dog, cat, panda, etc) will be $r$-separated for a value $r > 0$ depending on the image space (see Figure 1). In the next section, we empirically verify that this property actually holds for a number of standard image datasets.

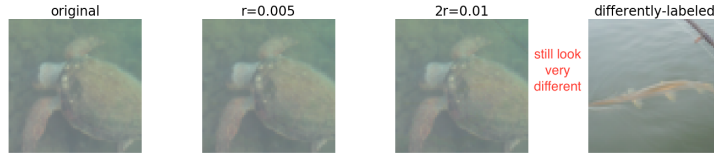

Figure 1: Intuitive example of $r$-separated images from Restricted ImageNet [57]. Even with a $2r$-perturbation, classes do not overlap. Moving $2r$ from turtle to fish still looks more like a turtle.

## 3   Real Image Datasets are $r$-Separated

We begin by addressing the question: Are image datasets $r$-separated for $\varepsilon \ll r$ and attack radii $\varepsilon$ in standard robustness experiments? While we cannot determine the underlying data distribution, we can empirically measure whether current training and test sets are $r$-separated. These measurements can potentially throw light on what can be achieved in terms of test robustness in real data.

We consider four datasets: MNIST, CIFAR-10, SVHN and Restricted ImageNet (ResImageNet), where ResImageNet contains images from a subset of ImageNet classes [32, 45, 57]. We present two statistics in Table 1 The *Train-Train Separation* is the $\ell_\infty$ distance between each training example and its closest neighbor with a different class label in the training set, while the *Test-Train Separation* is the $\ell_\infty$ distance between each test example and its closest example with a different class in the training set. See Figure 3 for histograms. We use exact nearest neighbor search to calculate distances. Table 1 also shows the typical adversarial attack radius $\varepsilon$ for the datasets; more details are in Appendix D.

|  | adversarial perturbation $\varepsilon$ | minimum Train-Train separation | minimum Test-Train separation |
|---|---|---|---|
| MNIST | 0.1 | 0.737 | 0.812 |
| CIFAR-10 | 0.031 | 0.212 | 0.220 |
| SVHN | 0.031 | 0.094 | 0.110 |
| ResImageNet | 0.005 | 0.180 | 0.224 |

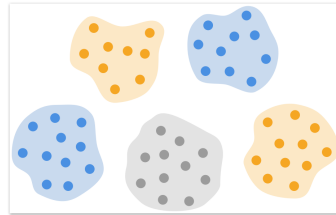

Table 1: Separation of real data is $3\times$ to $7\times$ typical perturbation radii.

Figure 2: Robust classifers exist if the perturbation is less than the separation.

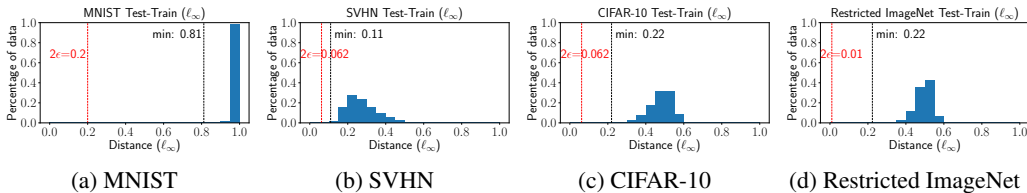

(a) MNIST      (b) SVHN      (c) CIFAR-10      (d) Restricted ImageNet

Figure 3: Train-Test separation histograms: MNIST, SVHN, CIFAR-10 and Restricted ImageNet.

Both the Train-Train and Test-Train separations are higher than $2\varepsilon$ for all four datasets. We note that SVHN contains a single duplicate example with multiple labels, and one highly noisy example; removing these three examples out of $73,257$ gives us a minimum Train-Train Separation of $0.094$, which is more than enough for attack radius $\varepsilon = 0.031 \approx 8/255$. Restricted ImageNet is similar with three pairs of duplicate examples, and two other highly noisy training examples (see Figures 7 and 8 in Appendix D). Barring a handful of highly noisy examples, real image datasets are indeed $r$-separated when $r$ is equal to the attack radii commonly used in adversarial robustness experiments.

These results imply that in real image data, the test images are far apart from training images from a different class. There perhaps are images of dogs which look like cats, but standard image datasets are quite clean, and such images mostly do not occur in either their test nor the training sets. In the next section, we explore consequences of this separation.

# 4  Robustness and Accuracy for $r$-Separated Data

We have just shown that four image datasets are indeed $r$-separated, for $\varepsilon \ll r$ where $\varepsilon$ is the typical adversarial perturbation used in experiments. We now show theoretically that if a data distribution is $r$-separated, then there exists a robust and accurate classifier that can be obtained by rounding a locally Lipschitz function. Additionally, we supplement these results in Appendix C by a constructive "existence proof" that demonstrates proof-of-concept neural networks with both high accuracy and robustness on some of these datasets; this illustrates that at least on these image datasets, these classifiers can potentially be achieved by neural networks.

## 4.1  $r$-Separation implies Robustness and Accuracy through Local Lipschitzness

We show that it is theoretically possible to achieve both robustness and accuracy for $r$-separated data. In particular, we exhibit a classifier based on a locally Lipschitz function, which has astuteness $1$ with radius $r$. Working directly in the multiclass case, our proof uses classifiers of the following form. If there are $C$ classes, we start with a vector-valued function $f : \mathcal{X} \to \mathbb{R}^C$ so that $f(\boldsymbol{x})$ is a $C$-dimensional real vector. We let $\mathsf{dist}(\boldsymbol{x}, \mathcal{X}^{(i)}) = \min_{\mathbf{z} \in \mathcal{X}^{(i)}} \mathsf{dist}(\boldsymbol{x}, \mathbf{z})$. We analyze the following function

$$f(\boldsymbol{x}) = \frac{1}{r} \cdot \left( \mathsf{dist}(\boldsymbol{x}, \mathcal{X}^{(1)}), \dots, \mathsf{dist}(\boldsymbol{x}, \mathcal{X}^{(C)}) \right). \tag{1}$$

In other words, we set $f(\boldsymbol{x})_i = \frac{1}{r} \cdot \mathsf{dist}(\boldsymbol{x}, \mathcal{X}^{(i)})$. Then, we define a classifier $g : \mathcal{X} \to [C]$ as

$$g(\boldsymbol{x}) = \operatorname*{argmin}_{i \in [C]} f(\boldsymbol{x})_i. \tag{2}$$

We show that accuracy and local Lipschitzness together imply astuteness (proofs in Appendix A).

**Lemma 4.1.** *Let $f : \mathcal{X} \to \mathbb{R}^C$ be a function, and consider $\boldsymbol{x} \in \mathcal{X}$ with true label $y \in [C]$. If*

- *$f$ is $\frac{1}{r}$-Locally Lipschitz in a radius $r$ around $\boldsymbol{x}$, and*
- *$f(\boldsymbol{x})_j - f(\boldsymbol{x})_y \geq 2$ for all $j \neq y$,*

*then $g(\boldsymbol{x}) = \operatorname{argmin}_i f(\boldsymbol{x})_i$ is astute at $\boldsymbol{x}$ with radius $r$.*

Finally, we show that there exists an astute classifier when the distribution is $r$-separated.

**Theorem 4.2.** *Suppose the data distribution $\mathcal{X}$ is $r$-separated, denoting $C$ classes $\mathcal{X}^{(1)}, \dots, \mathcal{X}^{(C)}$. There exists a function $f : \mathcal{X} \to \mathbb{R}^C$ such that*

(a) *$f$ is $\frac{1}{r}$-locally-Lipschitz in a ball of radius $r$ around each $\boldsymbol{x} \in \bigcup_{i \in [C]} \mathcal{X}^{(i)}$, and*

(b) *the classifier $g(\boldsymbol{x}) = \operatorname{argmin}_i f(\boldsymbol{x})_i$ has astuteness $1$ with radius $r$.*

While the classifier $g$ used in the proof of Theorem 4.2 resembles the 1-nearest-neighbor classifier, it is actually different on any finite sample, and the classifiers only coincide in the *infinite sample limit* or when the class supports are known.

**Binary case.** We also state results for the special case of binary classification. Let $\mathcal{X} = \mathcal{X}^+ \cup \mathcal{X}^-$ be the instance space with disjoint class supports $\mathcal{X}^+ \cap \mathcal{X}^- = \emptyset$. Then, we define $f : \mathcal{X} \to \mathbb{R}$ as

$$f(\boldsymbol{x}) = \frac{\mathsf{dist}(\boldsymbol{x}, \mathcal{X}^-) - \mathsf{dist}(\boldsymbol{x}, \mathcal{X}^+)}{2r}.$$

It is immediate that if $\mathcal{X}$ is $r$-separated, then $f$ is locally Lipschitz with constant $1/r$, and also the classifier $g = \mathsf{sign}(f)$ achieves the guarantees in the following theorem using the next lemma.

**Lemma 4.3.** *Let $f : \mathcal{X} \to \mathbb{R}$, and let $\boldsymbol{x} \in \mathcal{X}$ have label $y$. If (a) $f$ is $\frac{1}{r}$-Locally Lipschitz in a ball of radius $r > 0$ around $\boldsymbol{x}$, (b) $|f(\boldsymbol{x})| \geq 1$, and (c) $g(\boldsymbol{x})$ has the same sign as $y$, then the classifier $g = \mathsf{sign}(f)$ is astute at $\boldsymbol{x}$ with radius $r$.*

**Theorem 4.4.** *Suppose the data distribution $\mathcal{X} = \mathcal{X}^+ \cup \mathcal{X}^-$ is $r$-separated. Then, there exists a function $f$ such that (a) $f$ is $\frac{1}{r}$-locally Lipschitz in a ball of radius $r$ around all $\boldsymbol{x} \in \mathcal{X}$ and (b) the classifier $g = \mathsf{sign}(f)$ has astuteness 1 with radius $r$.*

A visualization of the function (and resulting classifier) from Theorem 4.4 for a binary classification dataset appears in Figure 4. Dark colors indicate high confidence (far from decision boundary) and lighter colors indicate the gradual change from one label to the next. The classifier $g = \mathsf{sign}(f)$ guaranteed by this theorem will predict the label based on which decision region (positive or negative) is closer to the input example. Figure 5 shows a pictorial example of why using a locally Lipschitz function can be just as expressive while also being robust.

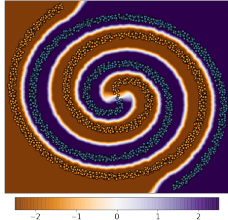

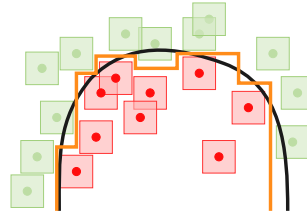

Figure 5: The classifier corresponding to the orange boundary has small local Lipschitz-ness because it does not change in the $\ell_\infty$ balls around data points. The black curve, however, is vulnerable to adversarial examples even though it has high clean accuracy.

Figure 4: Plot of $f(\boldsymbol{x})$ from Theorem 4.4 for the spiral dataset. The classifier $g = \mathsf{sign}(f)$ has high accuracy and astuteness because it gradually changes near the decision boundary.

## 4.2 Implications

We consider the consequences of Table 1 and Theorem 4.2 taken together. Then, in Section 5, we empirically explore the limitations of current robustness techniques.

**Significance for real data.** Theorem 4.2 refers to supports of distributions, while our measurements in Table 1 are on actual data. Hence, the results do not imply perfect *distributional* accuracy and robustness. However, our test set measurements suggest that even if the distributional supports may be close in the infinite sample limit, the close images are rare enough that we do not see them in the test sets. Thus, we still expect high accuracy and robustness on these test sets. Additionally, if we are willing to assume that the data supports are representative of the support of the distribution, then we can conclude the existence of a distributionally robust and accurate classifier. Combined with proof-of-concept results in Appendix C, we deduce that these classifiers can be implemented by neural networks. The remaining question is how such networks can be *trained* with existing methods.

**Optimally astute classifier and non-parametrics (comparison to Yang et al. [62]).** Prior work proposes adversarial pruning, a method that removes training examples until different classes are $r$-separated. They exhibit connections to maximally astute classifier, which they call the $r$-Optimal classifier for size $r$ perturbations [62]. Follow-up work proved that training various non-parametric classifiers after pruning leads them to converge to maximally astute classifiers under certain conditions [5]. Our result in Theorem 4.2 complements these efforts, showing that the $r$-Optimal classifier can be obtained by the classifier in Theorem 4.2. Moreover, we provide additional justification for adversarial pruning by presenting a new perspective on the role of data separation in robustness.

**Lower bounds on test error.** Our results also corroborate some recent works that use optimal transport to estimate a lower bound on the robust test accuracy of any classifier on standard datasets. They find that it is actually zero for typical perturbation sizes [4, 38]. In other words, we have further evidence that well-curated benchmark datasets are insufficient to demonstrate a tradeoff between robustness and accuracy, in contrast to predictions of an inevitable tradeoff [11, 12, 16, 57].

**Robustness is *not* inherently at odds with accuracy (comparison to Tsipras et al. [57]).** Prior work provides a theoretical example of a data distribution where any classifier with high test accuracy must also have small adversarial accuracy under $\ell_\infty$ perturbations. Their theorem led the authors to posit that (i) accuracy and robustness may be unachievable together due their inherently opposing goals, and (ii) the training algorithm may not be at fault [57]. We provide an alternative view.

Their distribution is defined using the following sampling process: the first feature is the binary class label (flipped with a small probability), and the other $d - 1$ features are sampled from one of two $(d - 1)$-dimensional Gaussian distributions either with mean $r$ or $-r$ depending on the true example label. While the means are separated with distance $2r$, their distribution is *not* $r$-separated due to the noise in the first feature combined with the infinite support of the Gaussians. Their lower bound is tight and only holds for $\ell_\infty$ perturbations $\varepsilon$ satisfying $\varepsilon \geq 2r$. Our experiments in Section 3 have already shown that $r$-separation is a realistic assumption, and typical perturbations $\varepsilon$ satisfy $\varepsilon \ll r$. Taken together with Theorem 4.2, we conclude that the robustness-accuracy tradeoff in neural networks and image classification tasks is *not intrinsic*.

# 5 A Closer Look at Existing Training Methods

So far we have shown that robustness and accuracy should both be achievable in principle, but practical networks continue to trade robustness off for accuracy. We next empirically investigate why this tradeoff might arise. One plausible reason might be that existing training methods do not impose local Lipschitzness properly; another may be that they do not generalize enough. We next explore these hypotheses in more detail, considering the following questions:

- How locally Lipschitz are the classifiers produced by existing training methods?
- How well do classifiers produced by existing training methods generalize?

These questions are considered in the context of one synthetic and four real datasets, as well as several plausible training methods for improving adversarial robustness. We do not aim to achieve best performance for any method, but rather to understand smoothness and generalization.

## 5.1 Experimental Methodology

We evaluate train/test accuracy, adversarial accuracy and local lipschitzness of neural networks trained using different methods. We also measure generalization gaps: the difference between train and test clean accuracy (or between train and test adversarial accuracy).

**Training Methods.** We consider neural networks trained via Natural training (Natural), Gradient Regularization (GR) [14], Locally Linear Regularization (LLR) [40], Adversarial Training (AT) [32], and TRADES [65]. Additionally, we use Robust Self Training (RST) [42], a recently introduced method that minimizes a linear combination of clean and robust accuracy in an attempt to improve robustness-accuracy tradeoffs. For fair comparison between methods, we use a version of RST that only uses labeled data. Both RST and TRADES have a parameter; for RST higher $\lambda$ means higher weight is given to the robust accuracy, while for TRADES higher $\beta$ means higher weight given to enforcing local Lipschitzness. Details are provided in Appendix B.

**Adversarial Attacks.** We evaluate robustness with two attacks. In this section, we use Projected gradient descent (PGD) [25] for adversarial accuracy with step size $\varepsilon/5$ and a total of 10 steps. The Multi-Targeted Attack (MT) [18] leads to similar conclusions; results in Appendix E.1.

**Measuring Local Lipschitzness.** For each classifier, we empirically measure the local Lipschitzness of the underlying function by the *empirical Lipschitz constant* defined as the following quantity

$$\frac{1}{n} \sum_{i=1}^{n} \max_{\boldsymbol{x}_i' \in \mathbb{B}_\infty(\boldsymbol{x}_i, \varepsilon)} \frac{\|f(\boldsymbol{x}_i) - f(\boldsymbol{x}_i')\|_1}{\|\boldsymbol{x}_i - \boldsymbol{x}_i'\|_\infty}. \tag{3}$$

A lower value of the empirical Lipschitz constant implies a smoother classifier. We estimate this through a PGD-like procedure, where we iteratively take a step towards the gradient direction $(\nabla_{\boldsymbol{x}_i'} \frac{\|f(\boldsymbol{x}_i) - f(\boldsymbol{x}_i')\|_1}{\|\boldsymbol{x}_i - \boldsymbol{x}_i'\|_\infty})$ where $\varepsilon$ is the perturbation radius. We use step size $\varepsilon/5$ and a total of 10 steps.

| architecture | CNN1 | | | | | | CNN2 | | | | | |
|---|---|---|---|---|---|---|---|---|---|---|---|---|
| | train acc. | test acc. | adv test acc. | test lipschitz | gap | adv gap | train acc. | test acc. | adv test acc. | test lipschitz | gap | adv gap |
| Natural | 100.00 | 99.20 | 59.83 | 67.25 | 0.80 | 0.45 | 100.00 | 99.51 | 86.01 | 23.06 | 0.49 | -0.28 |
| GR | 99.99 | 99.29 | 91.03 | 26.05 | 0.70 | 3.49 | 99.99 | 99.55 | 93.71 | 20.26 | 0.44 | 2.55 |
| LLR | 100.00 | 99.43 | 92.14 | 30.44 | 0.57 | 4.42 | 100.00 | 99.57 | 95.13 | 9.75 | 0.43 | 2.28 |
| AT | 99.98 | 99.31 | 97.21 | 8.84 | 0.67 | 2.67 | 99.98 | 99.48 | 98.03 | 6.09 | 0.50 | 1.92 |
| RST($\lambda$=.5) | 100.00 | 99.34 | 96.53 | 11.09 | 0.66 | 3.16 | 100.00 | 99.53 | 97.72 | 8.27 | 0.47 | 2.27 |
| RST($\lambda$=1) | 100.00 | 99.31 | 96.96 | 11.31 | 0.69 | 2.95 | 100.00 | 99.55 | 98.27 | 6.26 | 0.45 | 1.73 |
| RST($\lambda$=2) | 100.00 | 99.31 | 97.09 | 12.39 | 0.69 | 2.87 | 100.00 | 99.56 | 98.48 | 4.55 | 0.44 | 1.52 |
| TRADES($\beta$=1) | 99.81 | 99.26 | 96.60 | 9.69 | 0.55 | 2.10 | 99.96 | 99.58 | 98.10 | 4.74 | 0.38 | 1.70 |
| TRADES($\beta$=3) | 99.21 | 98.96 | 96.66 | 7.83 | 0.25 | 1.33 | 99.80 | 99.57 | 98.54 | 2.14 | 0.23 | 1.18 |
| TRADES($\beta$=6) | 97.50 | 97.54 | 93.68 | 2.87 | -0.04 | 0.37 | 99.61 | 99.59 | 98.73 | 1.36 | 0.02 | 0.80 |

Table 2: MNIST (perturbation 0.1). We compare two networks: CNN1 (smaller) and CNN2 (larger). We evaluate adversarial accuracy with the PGD-10 attack and compute Lipschitzness with Eq. (3). We also report the standard and adversarial generalization gaps.

| | CIFAR-10 | | | | | | Restricted ImageNet | | | | | |
|---|---|---|---|---|---|---|---|---|---|---|---|---|
| | train acc. | test acc. | adv test acc. | test lipschitz | gap | adv gap | train acc. | test acc. | adv test acc. | test lipschitz | gap | adv gap |
| Natural | 100.00 | 93.81 | 0.00 | 425.71 | 6.19 | 0.00 | 97.72 | 93.47 | 7.89 | 32228.51 | 4.25 | -0.46 |
| GR | 94.90 | 80.74 | 21.32 | 28.53 | 14.16 | 3.94 | 91.12 | 88.51 | 62.14 | 886.75 | 2.61 | 0.19 |
| LLR | 100.00 | 91.44 | 22.05 | 94.68 | 8.56 | 4.50 | 98.76 | 93.44 | 52.62 | 4795.66 | 5.32 | 0.22 |
| RST($\lambda$=.5) | 99.90 | 85.11 | 39.58 | 20.67 | 14.79 | 36.26 | 96.08 | 92.02 | 79.24 | 451.57 | 4.06 | 4.57 |
| RST($\lambda$=1) | 99.86 | 84.61 | 40.89 | 23.15 | 15.25 | 41.31 | 95.66 | 92.06 | 79.69 | 355.43 | 3.61 | 4.67 |
| RST($\lambda$=2) | 99.73 | 83.87 | 41.75 | 23.80 | 15.86 | 43.54 | 96.02 | 91.14 | 81.41 | 394.40 | 4.87 | 6.19 |
| AT | 99.84 | 83.51 | 43.51 | 26.23 | 16.33 | 49.94 | 96.22 | 90.33 | 82.25 | 287.97 | 5.90 | 8.23 |
| TRADES($\beta$=1) | 99.76 | 84.96 | 43.66 | 28.01 | 14.80 | 44.60 | 97.39 | 92.27 | 79.90 | 2144.66 | 5.13 | 6.66 |
| TRADES($\beta$=3) | 99.78 | 85.55 | 46.63 | 22.42 | 14.23 | 47.67 | 95.74 | 90.75 | 82.28 | 396.67 | 5.00 | 6.41 |
| TRADES($\beta$=6) | 98.93 | 84.46 | 48.58 | 13.05 | 14.47 | 42.65 | 93.34 | 88.92 | 82.13 | 200.90 | 4.42 | 5.31 |

Table 3: CIFAR-10 (perturbation 0.031) and Restricted ImageNet (perturbation 0.005). We evaluate adversarial accuracy with the PGD-10 attack and compute Lipschitzness with Eq. (3).

**Datasets.** We evaluate the various algorithms on one synthetic dataset: Staircase [41] and four real datasets: MNIST [26], SVHN [33], CIFAR-10 [24] and Restricted ImageNet [57]. We consider adversarial $\ell_\infty$ perturbations for all datasets. More details are in Appendix B.

## 5.2 Observations

Our experimental results, presented in Tables 2 and 3, provide a number of insights into the smoothness and generalization properties of classifiers trained by existing methods.

**How well do existing methods impose local Lipschitzness?** There is a large gap in the degree of local Lipschitzness in classifiers trained by AT, RST and TRADES and those trained by natural training, GR and LLR. Classifiers in the former group are considerably smoother than the latter. Classifiers produced by TRADES are the most locally Lipschitz overall, with smoothness improving with increasing $\beta$. AT and RST also produce classifiers of comparable smoothness – but less smooth than TRADES. Overall, local Lipschitzness appears mostly correlated with adversarial accuracy; the more robust methods are also the ones that impose the highest degree of local Lipschitzness. But there are diminishing returns in the correlation between robustness *and* accuracy and local Lipschitzness; for example, the local smoothness of TRADES improves with higher $\beta$; but increasing $\beta$ sometimes leads to drops in test accuracy even though the Lipschitz constant continues to decrease.

**How well do existing methods generalize?** We observe that for the methods that produce locally Lipschitz classifiers – namely, AT, TRADES and RST – also have large generalization gaps while natural training, GR and LLR generalize much better. In particular, there is a large gap between training and test accuracies of AT, RST and TRADES, and an even larger one between training and test adversarial accuracies. Although RST has better test accuracy than AT, it continues to have a large generalization gap with only labeled data. An interesting fact is that this generalization behaviour is quite unlike linear classification, where imposing local Lipschitzness leads to higher margin and better generalization [61] – imposing local Lipschitzness in neural networks, at least through these methods, appears to hurt generalization instead of helping. This suggests that these robust training methods may not be generalizing properly.

| | dropout | SVHN | | | | | CIFAR-10 | | | | |
|---|---|---|---|---|---|---|---|---|---|---|---|
| | | test acc. | adv test acc. | test lipschitz | gap | adv gap | test acc. | adv test acc. | test lipschitz | gap | adv gap |
| Natural | False | 95.85 | 2.66 | 149.82 | 4.15 | 0.87 | 93.81 | 0.00 | 425.71 | 6.19 | 0.00 |
| Natural | True | 96.66 | 1.52 | 152.38 | 3.34 | 1.22 | 93.87 | 0.00 | 384.48 | 6.13 | 0.00 |
| AT | False | 91.68 | 54.17 | 16.51 | 5.11 | 25.74 | 83.51 | 43.51 | 26.23 | 16.33 | 49.94 |
| AT | True | 93.05 | 57.90 | 11.68 | -0.14 | 6.48 | 85.20 | 43.07 | 31.59 | 14.51 | 44.05 |
| RST($\lambda$=2) | False | 92.39 | 51.39 | 23.17 | 6.86 | 36.02 | 83.87 | 41.75 | 23.80 | 15.86 | 43.54 |
| RST($\lambda$=2) | True | 95.19 | 55.22 | 17.59 | 1.90 | 11.30 | 85.49 | 40.24 | 34.45 | 14.00 | 33.07 |
| TRADES($\beta$=3) | False | 91.85 | 54.37 | 10.15 | 7.48 | 33.33 | 85.55 | 46.63 | 22.42 | 14.23 | 47.67 |
| TRADES($\beta$=3) | True | 94.00 | 62.41 | 4.99 | 0.48 | 7.91 | 86.43 | 49.01 | 14.69 | 12.59 | 35.03 |
| TRADES($\beta$=6) | False | 91.83 | 58.12 | 5.20 | 5.35 | 23.88 | 84.46 | 48.58 | 13.05 | 14.47 | 42.65 |
| TRADES($\beta$=6) | True | 93.46 | 63.24 | 3.30 | 0.45 | 5.97 | 84.69 | 52.32 | 8.13 | 11.91 | 26.49 |

Table 4: Dropout and generalization. SVHN (perturbation 0.031, dropout rate 0.5) and CIFAR-10 (perturbation 0.031, dropout rate 0.2). We evaluate adversarial accuracy with the PGD-10 attack and compute Lipschitzness with Eq. (3).

## 5.3 A Closer Look at Generalization

A natural follow-up question is whether the generalization gap of existing methods can be reduced by existing generalization-promoting methods in deep learning. In particular, we ask: *Can we improve the generalization gap of AT, RST and TRADES through generalization tools?*

To better understand this question, we consider two medium-sized datasets, SVHN and CIFAR-10, which nevertheless have a reasonably high gap between the test accuracy of the model produced by natural training and the best robust model. We then experiment with dropout [54], a standard and highly effective generalization method. For SVHN, we use a dropout rate of 0.5 and for CIFAR-10 a rate of 0.2. More experimental details are provided in the Appendix B.

Table 4 shows the results, contrasted with standard training. We observe that dropout narrows the generalization gap between training and test accuracy, as well as adversarial training and test accuracy significantly for all methods. For SVHN, after incorporating dropout, the best test accuracy is achieved by RST (95.19%) along with an adversarial test accuracy of 55.22%; the best adversarial test accuracy (62.41%) is with TRADES ($\beta = 3$) along with a test accuracy of (94.10%). Both accuracies are much closer to the accuracy of natural training (96.66%), and the test adversarial accuracies are also significantly higher. A similar narrowing of the generalization gap is visible for CIFAR-10 as well. Dropout also appears to make the networks smoother as test Lipschitzness also appears to improve for all algorithms for SVHN, and for TRADES for CIFAR-10.

**Dropout Improvements.** Our results show that the generalization gap of AT, RST and TRADES can be reduced by adding dropout; this reduction is particularly effective for RST and TRADES. Dropout additionally decreases the test local Lipschitzness of all methods – and hence promotes generalization all round – in accuracy, adversarial accuracy, and also local Lipschitzness. This suggests that combining dropout with the robust methods may be a good strategy for overall generalization.

## 5.4 Implications

Our experimental results lead to three major observations. We see that the training methods that produce the smoothest and most robust classifiers are AT, RST and TRADES. However, these robust methods also do not generalize well, and the generalization gap narrows when we add dropout.

**Comparison with Rice et al. [43].** An important implication of our results is that generalization is a particular challenge for existing robust methods. The fact that AT may sometimes overfit has been previously observed by [41, 43, 55]; in particular, Rice et al. [43] also experiments with a few generalization methods (but *not* dropout) and observes that only early stopping helps overcome overfitting to a certain degree. We expand the scope of these results to show that RST and TRADES also suffer from large generalization gaps, and that dropout can help narrow the gap in these two methods. Furthermore, we demonstrate that dropout often decreases the local Lipschitz constant.

# 6 Related Work

There is a large body of literature on developing adversarial attacks as well as defenses that apply to neural networks [7, 29, 56, 30, 32, 36, 35, 53, 58, 67]. While some of this work has noted that an increase in robustness is sometimes accompanied by a loss in accuracy, the phenomenon remains ill-understood. Raghunathan et al. [41] provides a synthetic problem where adversarial training overfits, which we take a closer look at in Appendix E. Raghunathan et al. [42] proposes the robust self training method that aims to improve the robustness and accuracy tradeoff; however, our experiments show that they do not completely close the gap particularly when using only labeled data.

Outside of neural networks, prior works suggest that lack of data may be responsible for low robustness [1, 5, 8, 9, 28, 48, 49, 59, 66]. For example, Schmidt et al. [48] provides an example of a linear classification problem where robustness in $\ell_\infty$ norm requires more samples than plain accuracy, and Wang et al. [59] shows that nearest neighbors would be more robust with more data. Some prior works also suggest that the robustness accuracy tradeoff may arise due to limitations of existing algorithms. Bubeck et al. [6] provides an example where finding a robust and accurate classifier is significantly more computationally challenging than finding one that is simply accurate, and Bhattacharjee and Chaudhuri [5] shows that certain non-parametric methods do not lead to robust classifiers in the large sample limit. It is also known that the Bayes optimal classifier is not robust in general [4, 44, 38, 57, 59], and it differs from the maximally astute classifier [5, 62].

Prior work has also shown a connection between adversarial robustness and local or global Lipschitzness. For linear classifiers, it is known that imposing Lipschitzness reduces to bounding the norm of the classifier, which in turn implies large margin [31, 61]. Thus, for linear classification of data that is linearly separable with a large margin, imposing Lipschitzness *helps* generalization.

For neural networks, Anil et al. [2], Qian and Wegman [39], Huster et al. [22] provide methods for imposing global Lipschitzness constraints; however, the state-of-the-art methods for training such networks do not lead to highly expressible functions. For local Lipschitzness, Hein and Andriushchenko [21] first showed a relationship between adversarial robustness of a classifier and local Lipschitzness of its underlying function. Following this, Weng et al. [60] provides an efficient way of calculating a lower bound on the local Lipschitzness coefficient. Many works consider a randomized notion of local smoothness, and they prove that enforcing it can lead to certifiably robust classifiers [27, 10, 37, 46].

# 7 Conclusion

Motivated by understanding when it is possible to achieve both accuracy and robustness, we take a closer look at robustness in image classification and make several observations. We show that many image datasets follow a natural separation property and that this implies the existence of a robust and perfectly accurate classifier that can be obtained by rounding a locally Lipschitz function. Thus in principle robustness and accuracy should both be achievable together on real world datasets.

We investigate the gap between theory and practice by examining the smoothness and generalization properties of existing training methods. Our results show that generalization may be a particular challenge in robust learning since all robust methods that produce locally smooth classifiers also suffer from fairly large generalization gaps. We then experiment with combining robust classification methods with dropout and see that this leads to a narrowing of the generalization gaps.

Our results suggest that the robustness-accuracy tradeoff in deep learning is not inherent, but it is rather a consequence of current methods for training robust networks. Future progress that ensures both robustness and accuracy may come from redesigning other aspects of the training process, such as customized optimization methods [3, 15, 25, 34, 43, 50, 52, 51] or better network architectures [13, 47, 69] in combination with loss functions that encourage adversarial robustness, generalization, and local Lipschitzness. Some recent evidence for improved network architectures has been obtained by Guo et. al. [19], who search for newer architectures with higher robustness from increased model capacity and feature denoising. A promising direction is to combine such efforts across the deep learning stack to reduce standard and adversarial generalization gaps.

## Broader Impact

In this paper we have investigated when it is possible to achieve both high accuracy and adversarial robustness on standard image classification datasets. Our motivation is partially to offer an alternative perspective to previous work that speculates on the inevitability of an accuracy-robustness tradeoff. In practice, if there were indeed a tradeoff, then robust machine learning technology is unlikely to be very useful. The vast majority of instances encountered by practical systems will be natural examples, whereas adversaries are few and far between. A self-driving car will mostly come across regular street signs and rarely come across adversarial ones. If increased robustness necessitates a loss in performance on natural examples, then the system's designer might be tempted to use a highly accurate classifier that is obtained through regular training and forgo robustness altogether. For adversarially robust machine learning to be widely adopted, accuracy needs to be achieved in conjunction with robustness.

While we have backed up our theoretical results by empirically verifying dataset separation, we are also ready to point out the many limitations of current robustness studies. The focus on curated benchmarks may lead to a false sense of security. Real life scenarios will likely involve much more complicated classification tasks. For example, the identification of criminal activity or the maneuvering of self-driving cars depend on a much broader notion of robustness than has been studied so far. Perturbations in $\ell_p$ distance cover only a small portion of the space of possible attacks.

Looking toward inherent biases, we observe that test accuracy is typically aggregated over all classes, and hence, it does not account for underrepresentation. For example, if a certain class makes up a negligible fraction of the dataset, then misclassifying these instances may be unnoticeable when we expect a drop in overall test accuracy. A more stringent objective would be to retain accuracy on each separate class, as well as being robust to targeted perturbations that may exploit dataset imbalance.

On a more positive note, we feel confident that developing a theoretically grounded discussion of robustness will encourage machine learning engineers to question the efficacy of various methods. As one of our contributions, we have shown that dataset separation guarantees the existence of an accurate and robust classifier. We believe that future work will develop new methods that achieve robustness by imposing both Lipschitzness and effective generalization. Overall, it is paramount to close the theory-practice gap by working on both sides, and we stand by our suggestion to further investigate the various deep learning components (architecture, loss function, training method, etc) that may compound the perceived gains in robustness and accuracy.

## Acknowledgments and Disclosure of Funding

Kamalika Chaudhuri and Yao-Yuan Yang thank NSF under CIF 1719133, CNS 1804829 and IIS 1617157 for support. Hongyang Zhang was supported in part by the Defense Advanced Research Projects Agency under cooperative agreement HR00112020003. The views expressed in this work do not necessarily reflect the position or the policy of the Government and no official endorsement should be inferred. Approved for public release; distribution is unlimited. This work was also supported in part by NSF IIS1763562 and ONR Grant N000141812861.

## Footnotes

[1]Code available at https://github.com/yangarbiter/robust-local-lipschitz.

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
