[Supplementary Material]

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

[2]CNN1 is retrieved from pytorch repository https://github.com/pytorch/examples/blob/master/mnist/main.py

[3]CNN2 is retrieved from TRADES [65] github repository https://github.com/yaodongyu/TRADES/blob/master/models/small_cnn.py

[4]https://github.com/pytorch/examples/blob/master/mnist/main.py

[5]https://github.com/yaodongyu/TRADES/blob/master/models/small_cnn.py

[6]https://github.com/cfinlay/tulip

[7]https://github.com/yaodongyu/TRADES

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

# A  Proofs for Astute Classifier Existence Results

*Proof of Lemma 4.1.* Suppose $\boldsymbol{x}' \in \mathcal{X}$ satisfies $\text{dist}(\boldsymbol{x}, \boldsymbol{x}') \leq r$. By the assumptions that $f$ is $\frac{1}{r}$-locally Lipschitz and $f(\boldsymbol{x})_j - f(\boldsymbol{x})_y \geq 2$, we have that

$$f(\boldsymbol{x}')_j \geq f(\boldsymbol{x})_j - 1 \geq f(\boldsymbol{x})_y + 1 \geq f(\boldsymbol{x}')_y,$$

where the first and third inequalities use Lipschitzness, and the middle inequality uses that

$$f(\boldsymbol{x})_j - f(\boldsymbol{x})_y \geq 2.$$

As this holds for all $j \neq y$, we have that

$$\operatorname*{argmin}_i f(\boldsymbol{x}')_i = \operatorname*{argmin}_i f(\boldsymbol{x})_i = y.$$

Thus, we see that $g(\boldsymbol{x}) = \operatorname{argmin}_i f(\boldsymbol{x})_i$ correctly classifies $\boldsymbol{x}$ while being astute with radius $r$.  □

*Proof of Theorem 4.2.* We first show that if the support of the distribution is $r$-separated, then there exists a function $f : \mathcal{X} \to \mathbb{R}^C$ satisfying:

1. If $\boldsymbol{x} \in \bigcup_{i \in [C]} \mathcal{X}^{(i)}$, then, $f(\boldsymbol{x})$ is $\frac{1}{r}$-locally-Lipschitz in a ball of radius $r$ around $\boldsymbol{x}$.

2. If $\boldsymbol{x} \in \mathcal{X}^{(y)}$, then $f(\boldsymbol{x})_j - f(\boldsymbol{x})_y \geq 2$ for all $j \neq y$.

Define the function

$$f(\boldsymbol{x}) = \frac{1}{r} \cdot \left( \text{dist}(\boldsymbol{x}, \mathcal{X}^{(1)}), \ldots, \text{dist}(\boldsymbol{x}, \mathcal{X}^{(C)}) \right).$$

In other words, we set $f(\boldsymbol{x})_i = \frac{1}{r} \cdot \text{dist}(\boldsymbol{x}, \mathcal{X}^{(i)})$. Then, for any $\boldsymbol{x}$, we have:

$$f(\boldsymbol{x})_i - f(\boldsymbol{x}')_i = \frac{\text{dist}(\boldsymbol{x}, \mathcal{X}^{(i)}) - \text{dist}(\boldsymbol{x}', \mathcal{X}^{(i)})}{r} \leq \frac{\text{dist}(\boldsymbol{x}, \boldsymbol{x}')}{r}$$

where we used the triangle inequality. This establishes (a). To establish (b), suppose without loss of generality that $\boldsymbol{x} \in \mathcal{X}^{(y)}$, which in particular implies that $f(\boldsymbol{x})_y = \text{dist}(\boldsymbol{x}, \mathcal{X}^{(y)}) = 0$. Then,

$$f(\boldsymbol{x})_j - f(\boldsymbol{x})_y = \frac{\text{dist}(\boldsymbol{x}, \mathcal{X}^{(j)})}{r} \geq \frac{\text{dist}(\mathcal{X}^{(y)}, \mathcal{X}^{(j)})}{r} \geq 2$$

because every pair of classes has distance at least $2r$ from the $r$-separated property.

Now observe that by construction, $f$ satisfies the two conditions in Lemma 4.1 at all $\boldsymbol{x} \in \bigcup_{i \in [C]} \mathcal{X}^{(i)}$. Thus, applying Lemma 4.1, we get that $g(\boldsymbol{x}) = \operatorname{argmin}_i f(\boldsymbol{x})_i$ has astuteness 1 with radius $r$ over any distribution over points in $\bigcup_{i \in [C]} \mathcal{X}^{(i)}$.  □

# B  Experimental Setup: More Details

Experiments run with NVIDIA GeForce RTX 2080 Ti GPUs. We report the number with a single run of experiment. The code for the experiments is available at https://github.com/yangarbiter/robust-local-lipschitz.

**Synthetic Staircase setup.** As a toy example, we first consider a synthetic regression dataset, which is known to show that adversarial training can seriously overfit when the sample size is too small [41]. We use the code provided by the authors to reproduce the result for natural training and AT, and we add results for GR, LLR, and TRADES. The model for this dataset is linear regression in a kernel space using cubic $B$-splines as the basis. Let $\mathcal{F}$ be the hypothesis set and the regularization term $\|f\|^2$ is the RKHS norm of the weight vector in the kernel space. The regularization term is set to $\lambda = 0.1$ and the result is evaluated using the mean squared error (MSE). For GR, we set $\beta = 10^{-4}$ and for LLR, we only use the local linearity $\gamma$ for regularization and the regularization strength is $10^{-2}$. The perturbation set $P(x, \varepsilon) = \{x - \varepsilon, x, x - \varepsilon\}$ considers only the point-wise perturbation.

**MNIST setup.** We use two different convolutional neural networks (CNN) with different capacity. The first CNN (CNN1) has two convolutional layers followed by two fully connected layer[2] and the second larger CNN (CNN2) has four convolutional layers followed by three fully connected layers[3]. We set the perturbation radius to 0.1. The network is optimized with SGD with momentum 0.9.

**SVHN setup.** We use the wide residual network WRN-40-10 [63] and set the perturbation radius to 0.031. The initial learning rate is set to 0.01 except LLR, AT and RST. We set the initial learning rate 0.001 for them. The network is optimized with SGD without momentum (the default setting in `pytorch`).

**CIFAR10 setup.** Following [32, 65], we use the wide residual network WRN-40-10 [63] and set the perturbation radius to 0.031. The initial learning rate is set to 0.01 except RST. We set the initial learning rate 0.001 for them. Data augmentation is performed. When performing data augmentation, we randomly crop the image to $32 \times 32$ with 4 pixels of padding then perform random horizontal flips. The network is optimized with SGD without momentum (the default setting in `pytorch`).

**Restricted ImageNet setup.** Following [57], we set the perturbation radius $\varepsilon = 0.005$, use the residual network (ResNet50) [20] and use Adam [23] to optimize. Data augmentation is performed: During training, we resize an image to $72 \times 72$ and randomly crop to $64 \times 64$ with 8 pixels padding. When evaluating, we resize the image to $72 \times 72$ and crop in the center resulting in a $64 \times 64$ image.

| dataset | MNIST | SVHN | CIFAR10 | Restricted ImageNet |
|---|---|---|---|---|
| network structure | CNN1 / CNN2 | WRN-40-10 | WRN-40-10 | ResNet50 |
| optimizer | SGD | SGD | SGD | Adam |
| batch size | 64 | 64 | 64 | 128 |
| perturbation radius | 0.1 | 0.031 | 0.031 | 0.005 |
| perturbation step size | 0.02 | 0.0062 | 0.0062 | 0.001 |
| # train examples | 60000 | 73257 | 50000 | 257748 |
| # test examples | 10000 | 26032 | 10000 | 10150 |
| # classes | 10 | 10 | 10 | 9 |

Table 5: Experimental setup and parameters for the four real datasets that we test on in this paper. No weight decay is applied to the model.

**Details on the network structure.**

- CNN1 is retrieved from pytorch repository.[4]
- CNN2 is retrieved from TRADES [65] github repository.[5]
- WRN-40-10 represents the wide residual network [63] with depth equals to forty and widen factor equals to ten.
- ResNet50 represents the residual network with 50 layers [20].

**Learning rate schedulers for each dataset**

- MNIST: We run 160 epochs on the training dataset, where we decay the learning rate by a factor 0.1 in the 40th, 80th 120th and 140th epochs.
- SVHN: We run 60 epochs on the training dataset, where we decay the learning rate by a factor 0.1 in the 30th and 50th epochs.
- CIFAR10: We run 120 epochs on the training dataset, where we decay the learning rate by a factor 0.1 in the 40th, 80th and 100th epochs.
- Restricted ImageNet: We run 70 epochs on the training dataset, where we decay the learning rate by a factor 0.1 in the 40th and 60th epochs.

### B.0.1 Spiral Dataset

Here we provide the details for generating the spiral dataset in Figure 4.

We take $x$ as a uniform sample $[0, 4.33\pi]$, the noise level is set to $0.75$ (uniform $[0, 0.75]$).

We construct the negative examples using the transform:

$$(-x\cos x + uniform(noise), x\sin x + uniform(noise))$$

We construct the positive examples using the transform:

$$(-x\cos x + uniform(noise), -x\sin x + uniform(noise))$$

### B.0.2 Details on the baseline algorithms

**Gradient Regularization (GR).** The Gradient Regularization (GR) is in the form of soft regularization. We use the latest work by Finlay and Oberman [14] for our experiments. In general, GR models can be formulated as adding a regularization term on the norm of gradient of the loss function:

$$\min_f \mathbb{E}\Big\{ \mathcal{L}(f(\mathbf{X}), Y) + \beta \|\nabla_{\mathbf{X}}\mathcal{L}(f(\mathbf{X}), Y)\|_2^2 \Big\}.$$

Finlay and Oberman [14] compute the gradient term through a finite difference approximation. Let $d = \frac{\nabla f(\mathbf{X})}{\|\nabla f(\mathbf{X})\|_2}$ and $h$ be the step size. Then,

$$\|\nabla f(\mathbf{X})\|_2^2 \approx \left( \frac{\mathcal{L}(f(\mathbf{X}+hd), Y) - \mathcal{L}(f(\mathbf{X}), Y)}{h} \right)$$

We use the publicly available implementation.[6]

**Locally-Linear Regularization model (LLR).** Qin et al. [40] propose to regularize the local linearity through the motivation that AT with PGD increases the model's local linearity. The authors first formulate the function $g$ to evaluate the local linearity of a model.

$$g(f, \delta, \mathbf{X}) = |\mathcal{L}(f(\mathbf{X}+\delta), Y) - \mathcal{L}(f(\mathbf{X}), Y) - \delta^T \nabla_{\mathbf{X}}\mathcal{L}(f(\mathbf{X}), Y)|$$

Define $\gamma(\varepsilon, \mathbf{X}) = \mathbb{E}\Big\{ \max_{\delta \in B(\mathbf{X}, \varepsilon)} g(f, \delta, \mathbf{X}) \Big\}$ and also $\delta_{LLR} = \mathbb{E}\Big\{ \text{argmax}_{\delta \in B(\mathbf{X}, \varepsilon)} g(f, \delta, \mathbf{X}) \Big\}$. The loss function for Locally-Linear Regularization (LLR) model is

$$\mathbb{E}\Big\{ \mathcal{L}(f(\mathbf{X}), Y) + \lambda\gamma(\varepsilon, \mathbf{X}) + \mu\|\delta_{LLR}^T \nabla_{\mathbf{X}}\mathcal{L}(f(\mathbf{X}), Y)\| \Big\}$$

We use our own implementation of LLR.

**Adversarial training (AT).** Adversarial training is a successful defense by Madry et al. [32] that trains based on adversarial examples:

$$\min_f \mathbb{E}\Big\{ \max_{\mathbf{X}' \in \mathbb{B}(\mathbf{X}, \varepsilon)} \mathcal{L}(f(\mathbf{X}'), Y) \Big\}. \tag{4}$$

**Robust self-training (RST).** Robust self-training is a defense proposed by Raghunathan et al. [42] to improve the tradeoff between standard and robust error that occurs with AT. The RST in the original paper includes the use unlabeled data. However, to be fair in the experiments, we considers only the supervised learning part. RST uses the following optimization problem for the loss function:

$$\min_f \mathbb{E}\Big\{ \mathcal{L}(f(\mathbf{X}), Y) + \beta \max_{\mathbf{X}' \in \mathbb{B}(\mathbf{X}, \varepsilon)} \mathcal{L}(f(\mathbf{X}'), Y) \Big\}.$$

**Locally-Lipschitz models (TRADES).** One of the best methods for robustness via smoothness is TRADES [65], which has been shown to obtain state-of-the-art adversarially accuracy in many cases. TRADES uses the following optimization problem for the loss function:

$$\min_f \mathbb{E}\Big\{ \mathcal{L}(f(\mathbf{X}), Y) + \beta \max_{\mathbf{X}' \in \mathbb{B}(\mathbf{X}, \varepsilon)} \mathcal{L}(f(\mathbf{X}), f(\mathbf{X}')) \Big\},$$

where the second term encourages local Lipschitzness. We use the publicly available implementation.[7]

## C   Proof-of-Concept Classifier

Our theoretical result in Theorem 4.2 leaves two concerns about practical solutions: (i) we need to verify that existing networks can achieve high robustness and accuracy, and (ii) we need training methods to converge to such solutions. For the first concern, we present proof-of-concept networks that use standard architectures, but they have an unfair advantage: they can use the test data. While this may seem unreasonable, we argue that the results in Table 1 provide multiple insights. This process is sufficient to establish the *existence* of a robust and accurate classifier. With access to the test data, current training methods can plausibly train a nearly-optimal robust classifier (we use robust self-training with $\lambda$=2). Finally, the robust accuracies can actually get close to 100% on MNIST, SVHN, and CIFAR-10. These insights reinforce our claim that robustness and accuracy are both achievable by neural networks on image classification tasks.

|  | perturbation $\varepsilon$ | accuracy | adversarial accuracy |
|---|---|---|---|
| MNIST | 0.1 | 99.99 | 99.98 |
| SVHN | 0.031 | 100.00 | 99.90 |
| CIFAR-10 | 0.031 | 100.00 | 99.99 |

Table 6: Proof-of-concept: demonstrating a robust network trained with access to the test set.

|  | MNIST | SVHN | CIFAR10 |
|---|---|---|---|
| network structure | CNN2 | WRN-40-10 | WRN-40-10 |
| optimizer | SGD | Adam | Adam |
| batch size | 128 | 64 | 64 |
| epochs | 40 | 60 | 60 |
| perturbation radius | 0.1 | 0.031 | 0.031 |
| perturbation step size | 0.02 | 0.0062 | 0.0062 |
| initial learning rate | 0.0001 | 0.001 | 0.01 |

Table 7: Setup for the Proof-of-Concept classifiers.

## D   Separation Experiment Results

Figures 6 shows the distribution of the train-train and test train separation for each dataset.

**Random label.**   In addition, we conducted a random label experiment to show that in regular datasets, the distance between same-class examples are smaller than differently-labeled examples. Table 8 shows the minimum and average separation for randomly labeled and natural dataset. Figure 9 plots the minimum separation for these two dataset and clearly shows that natural dataset is more well-separated than random-labeled dataset.

|  | $\varepsilon$ | randomly labeled | | | | original labels | | | |
|---|---|---|---|---|---|---|---|---|---|
|  |  | train-train | | test-train | | train-train | | test-train | |
|  |  | min | mean | min | mean | min | mean | min | mean |
| MNIST | 0.100 | 0.231 | 0.902 | 0.290 | 0.904 | 0.737 | 0.990 | 0.812 | 0.990 |
| CIFAR-10 | 0.031 | 0.125 | 0.476 | 0.098 | 0.475 | 0.212 | 0.479 | 0.220 | 0.479 |
| SVHN | 0.031 | 0.012 | 0.259 | 0.102 | 0.271 | 0.094 | 0.264 | 0.110 | 0.274 |
| ResImageNet | 0.005 | 0.000 | 0.485 | 0.000 | 0.483 | 0.180 | 0.492 | 0.224 | 0.492 |

Table 8: Separation results on real datasets for both original labels and randomly assigned labels.

(a) MNIST train                       (b) SVHN train

(c) CIFAR-10 train                 (d) Restricted ImageNet train

Figure 6: Train-Train separation histograms: MNIST, SVHN, CIFAR-10 and Restricted ImageNet.

(a)                       (b)                       (c)

Figure 7: Images ignored in SVHN. The left most image appeared twice in the training set and one labeled as a one and the other one labeled as a five. The other two images are the closest images to each other in $\ell_\infty$ distance. The middle one is labeled as a five and the right most one is labeled as a one (which is clearly miss labeled).

(a)                       (b)                       (c)

Figure 8: Images ignored in Restricted ImageNet. These three images appeared twice in the training set and labeled differently (one labeled as a cat and the other one labeled as a dog).

Figure 9: Separation results for four image datasets. We measure the separation for the original labels, and we also perform the experiment where we randomly label the test and train examples. We see that in MNIST, CIFAR-10, and ResImageNet, the separation diminishes quite a bit when using random labels. Indeed for ResImageNet, there are a number of duplicated examples that appear multiple times in the dataset. Overall, we conclude that the separation is much larger between *different* classes, while this is not the case *within* the same class.

# E Further Experimental Results

## E.1 Multi-targeted Attack Results

Certain prior works have suggested that the multi-targeted (MT) attack [18] is stronger than PGD. For example, the MT attack is highlighted as a selling point for LLR [40]. For completeness, we complement our empirical results from earlier by running all of the experiments using the MT attack. We run MT attack with 20 iterations for each target. Tables 9 to 14 provide the results.

We verify that our discussion about accuracy, robustness, and Lipschitzness remains valid using this attack. Comparing with the results using the PGD attack (Tables 2 to 4 above), the results with the MT attack gives a slightly lower adversarial test accuracy for all methods. The drop in accuracy is usually around 1–5%. This is within our expectation as this attack is regarded as a stronger attack than PGD.

The MT results still justify the previous discussion from Section 5 in general. Training methods leading to models with higher adversarial test accuracy are more locally smooth (smaller local Lipschitz constant during testing). Overall, we believe that seeing consistent results between PGD and MT only strengthens our argument that robustness requires some local Lipschitzness, and moreover, that the accuracy-robustness trade-off may not be necessary for separated data.