[Reviews · NeurIPS 2020]

Review 1

Summary and Contributions: The paper examines robustness-accuracy tradeoffs, extending previous work of Chaudhari’s group on robustness (measured by astuteness) and local smoothness examined by multiple groups. The theoretical results are fairly restrictive, but the empirical results add further evidence to insights in the literature.

Strengths: Under the assumption that classes are perfectly separable, with a minimum distance 2r between clusters, the paper proves that there exists a classifier that, with probability one, is impervious to attacks of magnitude r. The theoretical results are rigorously established. The evaluation section provides multiple insights or adds further evidence to insights already in the literature. The paper presents and tests two hypotheses: local smoothness improves robustness; the empirical Lispschitz constant of several trained NNs is computed; smoother classifiers (AR, RST, TRADES) are shown empirically to be more robust, but to also have larger generalization gap (difference between train and test accuracies); the paper then observes that dropout reduces the gap. In this sense, the paper adds some insights to the robust-accurate-generalizable trade off. Nonetheless, the 1/r locally-Lispschitz condition in the Theorem (and subsequent empirical tests) does provide an analytical characterization of the classifier. The proof considers a classifier that essentially outputs the label of the nearest class (in essence, this involves computing the distance between the test point and every point in the training set, in the data domain, similar in principle to the NN-1 classifier). In principle, this means that the entire training data must be available at test time, which is typically infeasible. On the other hand, the NN itself would not be needed at test time, since the labels for the training data are already known. The paper presents and tests two hypotheses: local smoothness improves robustness; the empirical Lispschitz constant of several trained NNs is computed; smoother classifiers (AR, RST, TRADES) are shown empirically to be more robust, but to also have larger generalization gap (difference between train and test accuracies); the paper then observes that dropout reduces the gap. In this sense, the paper adds some insights to the robust-accurate-generalizable trade off. The statement on L160, “well-curated benchmark datasets are insufficient to demonstrate a tradeoff between robustness and accuracy” offers a good warning to the community. Overall a well-written paper.

Weaknesses: My issues with the paper are twofold: 1) There exist no real world data where the classes are well-separated. Most real-world data are not curated; images suffer from different lighting and lens conditions; real images of objects to not occur in isolation, i.e. objects are always obscured by other objects. Indeed, the authors make statements to this effect under Broader Impact. 2) The classifier proposed in the existence proof essentially computes the distance of a test point to every point in the training data. Thus all of the training data must be available at test time. In practice, this would be infeasible, or contrary to policy.

Correctness: Yes

Clarity: Very well written.

Relation to Prior Work: Yes, literature review is thorough, and good positioning of the work in the context of SotA.

Reproducibility: Yes

Additional Feedback: The assumption of r-separability is strong, and unlikely to hold in practice. The proposed classifier requires that the entire training data be available at inference time. This is unrealistic. ** Added post-discussion ** The authors' response clarifies some but not all of my concerns. I appreciate the added empirical results, and the clarification of the intent to provide theory not necc. deployable algorithms. I have increased my overall score.


Review 2

Summary and Contributions: The paper argues that there is no intrinsic tradeoff between robustness and accuracy on standard image data, but empirically, even with a small Lipschitz constant (which implies the existence of an astute classifier), we encounter the tradeoff because of a poor generalization of current state-of-the-art robust training methods. In addition, the authors suggest to use dropout to achieve a better generalization performance addition to the robust training methods.

Strengths: The paper has its strengths on significance and novelty of the contribution. The paper clarifies the point that the robust-accuracy tradeoff is not intrinsic, while Tsipras et al. 2019 and TRADES constructed toy data on which we must sacrifice either one of the two, accuracy and robustness.

Weaknesses: The authors empirically demonstrated that robust training methods covered in the paper appear to hurt generalization. More importantly, however, the discussion on generalization is insufficient and lacks theoretical evidence at a NeurIPS level. -----------------------after rebuttal I read the Appendix for the experiments against Multi-Targeted attack. I agree that it is well-evaluated against strong enough attack. However I still believe the discussion is weak. I would like to keep my decision, but it is a borderline decision. It should be compared to Early Stopping or other methods found to be not successful in Rice et al.

Correctness: The claims of the paper and measuring local Lipschitzness appear to be correct. However, empirical methodology used to evaluate the robustness is not appropriate. It only tested against the PGD-10 attack which cannot represent the true robustness.

Clarity: The paper is well written.

Relation to Prior Work: As mentioned before, the paper clarifies the point that the robust-accuracy tradeoff is not intrinsic, while previous work mainly argued that the tradeoff may exist in a certain type of data.

Reproducibility: Yes

Additional Feedback:


Review 3

Summary and Contributions: In this paper, the authors showed that four image datasets (MNIST, CIFAR10, SVHN, ResImageNet) can be separated through empirical measurements and the distances between the samples of different classes are obviously larger than the perturbation radius epsilon. With this separated data distribution, the authors theoretically showed that there exists a classifier which can achieve both the robustness and accuracy at the same time. It can be achieved by rounding a locally Lipschitz function. The authors also explored the smoothness and the generalization properties of the current training methods such as AT, TRADES, and RST, showing that local Lipschitzness appears mostly correlated with adversarial accuracy. The authors also showed the positive impact of dropout on narrowing the generalization gaps and sometimes making the classifiers smoother.

Strengths: The experiments on the four datasets are interesting, showing real image datasets are separated. Under this scenario, in some ways the theory proposed by authors are against the previous works that a robustness-accuracy tradeoff may be inevitable. The theoretical proof that a both robust and accuracy classifier exists is interesting and it may open up a new direction.

Weaknesses: The paper claimed and showed the robustness and accuracy are achievable at the same time for the real image datasets. This is good though there are already a lot of papers showing that adversarial examples can actually boost the performance when the perturbation is small. So the conclusion or the claim is not very new at least to the reviewer. On the other hand, it is somehow unfortunately that the authors did not propose a framework on how to attain this target of encouraging both an accuracy increase as well as robustness (Dropout is not a solution). In other words, we could not see both a robust and accurate frame from the Experiments in Section5. The authors claimed “Overall, the local Lipschitzness appears mostly correlated with adversarial accuracy; more robust methods are also the ones that impose the highest degree of local Lipschitzness.” in Line 218 and 219. However, I note several inconsistencies against this claim in Table 2 and Table 3, such as “TRADES(β=6) in CNN1 in Table2” which has the highest degree of local Lipschitzness but lower adv test accuracy than others, “TRADES(β=1) in Restricted ImageNet in Table3”which has lower degree but higher adv test accuracy than “RST(λ=1), RST(λ=.5), GR”. These make the experimental results not solid/unpersuasive to me, or the authors should explain and analyze more about such“special cases” against their conclusions. Moreover, the authors should better design additional experiments to support their theory. The theory could be more solid and meaningful if there are more analysis and explanations about why the current frameworks failed to achieve robustness and high natural accuracy.

Correctness: I checked the most derivations, theory and I assessed the sensibility of the experiments.

Clarity: The paper is well written and I found that it is easy to me to follow their theoretical setting.

Relation to Prior Work: The authors generally did a good job on discussing previous contributions in Section 4.2 and 5.4.

Reproducibility: Yes

Additional Feedback: As stated in Weakness, the authors may better try to explain and analyze the experimental results in Table 2 and Table 3. Some of the results showed that the local Lipschitzness is less correlative with the adversarial accuracy and standard accuracy, which makes the theory seem non-solid/unpersuasive. In other words, the authors should better design more experiments to support their theory further. As shown in Table 4, “dropout decreases the test local Lipschitzness of all the methods and promotes generalization all round” and the authors discussed reference [43] about early stopping. It would be better to compare the dropout and early stopping. =======after the rebuttal====== I read carefully the rebuttal. Unfortunately, I am still not very convinced that I should increase my rating. I appreciated the authors rebuttal, which help clarify some points though many of my concerns were not addressed. Nonetheless, in the reviewer's personal opinion, the contribution of the paper may not be very significant. Again, there are already some existing works showing that robustness and accuracy can be achieved and the paper showed "that the robustness and accuracy should be nearly perfect for standard datasets". The work may provide some evidence to the question about "robustness and accuracy", it may still need further discussion or solid explorations to enhance the significance of the paper. I would keep my rating but would leave the decision to the AC or PC chair.

[Author Response · NeurIPS 2020]

We thank all reviewers for their comments. We are glad everyone found out paper well written. Our main contribution
is conceptual, showing that it is possible to achieve both robustness and accuracy in principle (contrary to previous
work). We also identify that the robustness-accuracy gap seems to be caused by generalization shortcomings, which
highlights the importance of addressing this with better algorithms in the future. Below we address specific comments.

**R1.** *"There exist no real world data where the classes are well-separated... images suffer from different lighting and*
*lens conditions... objects do not occur in isolation..."*
It seems like there is some confusion between classes being *well-separated* and the notion we use, *$r$-separation*. The
property of $r$-separation merely requires two images with different classes to have distance at least $r$. If the lighting
changes, then this may increase the in-class variation, but it is **not** going to make the image look like it is from another
class (in pixel space). To be concrete, here is a turtle and a fish from Restricted ImageNet. A perturbation distance
$r = 0.005$ is used for this dataset (the separation, in terms of $r$-separation, is much larger). Even with a $2r$-perturbation
(for a dim image), the classes do not overlap. Moving $2r$ from the turtle to the fish **still** looks much more like a turtle.

original     r=0.005     2r=0.01     differently-labeled

still look very different

In Section 3, we show that four datasets (MNIST, CIFAR-10, SVHN, ResImageNet) are all $r$-separated (but, of course,
they are not highly clustered). Even in these $r$-separated datasets, we see a robustness-accuracy tradeoff, which implies
that **the problem is with the algorithms**. If robustness is needed for more overlapping datasets, then one option is to
make them more separated (at the cost of some accuracy) using Adversarial Pruning from Yang et al., 2020.

**R1.** *"The classifier proposed in the existence proof essentially computes the distance of a test point to every point in the*
*training data. ... In practice, this would be infeasible, or contrary to policy."*
We **agree** that there are limitations on the classifier used for the existence proof (robustness and accuracy in practice is a
huge open problem). However, our theory result is not intended to be used in practice. Instead, it rigorously establishes
that the robustness-accuracy trade-off is *not intrinsic* for $r$-separated data.

**R2.** *"...empirically demonstrated that robust training methods covered in the paper appear to hurt generalization. More*
*importantly, however, the discussion on generalization is insufficient and lacks theoretical evidence at a NeurIPS level."*
The gap between training and testing accuracies are larger for the robust methods. A theory of generalization for neural
networks that can explain adversarial examples adequately is a **long-standing open problem**. Some recent[1] works[2]
claim to do so, but they are still in their infancy do not explain many aspects of the problem.

**R2.** *"It only tested against the PGD-10 attack which cannot represent the true robustness."*
Actually, in Section E.1 of our submission we test against the **multi-targeted attack** for a total of 200 iterations for
a dataset with 10 classes (this is essentially SOTA for $\ell_\infty$) and we see that the overall trend between adversarial test
accuracy and local Lipschitzness remains the same (even though the adv. accuracy goes down across the board).

**R3.** *" ..., it is somehow unfortunately that the authors did not propose a framework on how to attain this target of*
*encouraging both an accuracy increase as well as robustness ... "*
Two long-standing open problems in this field are: (i) understand theoretically how networks generalize adversarially,
(ii) design a framework that achieves robustness and accuracy empirically. In our work, we do not claim to design such
framework. Instead, one of the main contributions of our work is to show evidence that **such a framework should**
**exist**, especially for standard image datasets (this was not clear before). The result encourages future work to focus on
finding such framework instead of believing that robustness and accuracy are at odds with each other.

**R3.** *The paper claimed and showed the robustness and accuracy are achievable at the same time for the real image*
*datasets. ... So the conclusion or the claim is not very new at least to the reviewer.*
The reviewer seems to be conflating two results that we believe are quite different.We are showing that the robustness
and accuracy should be *nearly perfect* for standard datasets. This is **not evidenced** by the recent works showing that
robustness can help accuracy a little bit. For example, on CIFAR-10 or ResImageNet, many papers report trade-offs, and
there is little evidence of any current solutions that get close to what we can show in theory. Therefore, we think that our
theoretical results actually tell quite a new story about **why** the trade-off is not intrinsic, and about why generalization
seems to be the key barrier.

## Footnotes

[1] Belkin, Mikhail, Daniel J. Hsu, and Partha Mitra. "Overfitting or perfect fitting? risk bounds for classification and regression rules that interpolate.", 2018.

[2] Wei, Colin, and Tengyu Ma. "Improved sample complexities for deep networks and robust classification via an all-layer margin.", 2019.


[Meta-Review · NeurIPS 2020]

The paper provides novel and solid contributions, both from the theoretical and the empirical standpoints. Its main message is that robustness to adversarial attacks and accuracy are not necessarily contradictory and can be achieved at the same time. To illustrate this claim, the authors observe that many machine learning problems exhibit a natural separation of data which is larger than the size of adversarial attacks. They also demonstrate and prove that smoothness can be used for ensuring both accuracy and robustness. The particular implementation of the smoothness criterion proposed in this paper received some criticism in reviews but this would hopefully motivate authors and other researchers to investigate alternative methods for ensuring smoothness of the decision functions.